# Epilepsy and nodding syndrome in association with an *Onchocerca volvulus* infection drive distinct immune profile patterns

**Kathrin Arndts**[1‡]*, **Josua Kegele**[2‡], **Alain S. Massarani**[1], **Manuel Ritter**[1], **Thomas Wagner**[3,4,5], **Kenneth Pfarr**[1,4], **Christine Lämmer**[1,4], **Peter Dörmann**[6], **Helga Peisker**[6], **Dirk Menche**[7], **Mazen Al-Bahra**[7], **Clarissa Prazeres da Costa**[5,8,9], **Erich Schmutzhard**[10], **William Matuja**[11], **Achim Hoerauf**[1,4,12‡]*, **Laura E. Layland-Heni**[1,4‡], **Andrea S. Winkler**[9,13,14,15‡]*

1 Institute of Medical Microbiology, Immunology and Parasitology (IMMIP), University Hospital Bonn (UKB), Bonn, German, 2 Hertie Institute for Clinical Brain Research, Department of Neurology and Epileptology, University of Tübingen, Tübingen, Germany, 3 Center for Pediatric and Adolescent Medicine, University Heidelberg, Heidelberg, Germany, 4 German Center for Infection Research (DZIF), Partner Site Bonn-Cologne, Bonn, Germany, 5 German Center for Infection Research (DZIF), Partner Site Munich, Germany, 6 Institute of Molecular Physiology and Biotechnology of Plants, University of Bonn, Bonn, Germany, 7 Kekulé Institute of Organic Chemistry and Biochemistry, University of Bonn, Bonn, Germany, 8 Institute of Medical Microbiology, Immunology and Hygiene, Technical University of Munich, Munich, Germany, 9 Center for Global Health, School of Medicine, Technical University of Munich, Munich, Germany, 10 Department of Neurology, Medical University Innsbruck, Innsbruck, Austria, 11 Department of Neurology, Muhimbili University of Health and Allied Sciences, Dar es Salaam, Tanzania, 12 German-West African Centre for Global Health and Pandemic Prevention (G-WAC), Partner Site Bonn, Bonn, Germany, 13 Department of Neurology, Technical University of Munich, Munich, Germany, 14 Department of Community Medicine and Global Health, Institute of Health and Society, University of Oslo, Oslo, Norway, 15 Department of Global Health and Social Medicine, Harvard Medical School, Boston, Massachusetts, USA

‡ KA and JK share first authorship on this work. AH, LEL-H and ASW share last authorship on this work.
* kathrin.arndts@ukbonn.de (KA); hoerauf@uni-bonn.de (AH); andrea.winkler@tum.de (ASW)

**Data Availability Statement:** All relevant data are within the paper and its Supporting Information

## Abstract

Previous studies have described the association of onchocerciasis (caused by *Onchocerca volvulus*) with epilepsy, including nodding syndrome, although a clear etiological link is still missing. Cases are found in different African countries (Tanzania, South Sudan, Uganda, Democratic Republic of the Congo, Central African Republic and Cameroon). In our study we investigated immunological parameters (cytokine, chemokine, immunoglobulin levels) in individuals from the Mahenge area, Tanzania, presenting with either epilepsy or nodding syndrome with or without *O. volvulus* infection and compared them to *O. volvulus* negative individuals from the same endemic area lacking neurological disorders. Additionally, cell differentiation was performed using blood smears and systemic levels of neurodegeneration markers, leiomodin-1 and N-acetyltyramine-O, β-glucuronide (NATOG) were determined. Our findings revealed that cytokines, most chemokines and neurodegeneration markers were comparable between both groups presenting with epilepsy or nodding syndrome. However, we observed elevated eosinophil percentages within the *O. volvulus* positive epilepsy/nodding syndrome patients accompanied with increased eosinophilic cationic protein (ECP) and antigen-specific IgG levels in comparison to those without an *O. volvulus*

files. All data underlying the findings described in this manuscript are available in S4 Table.

**Funding:** JK was funded by the Deutsche Gesellschaft für Epileptologie (DGFE). The clinical leave stipend of TW was supported by a grant from the German Center of Infectious Disease (DZIF; Verbundprojekt: TI 07.001) of which ASW, CP and AH were co-applicants. In addition, AH, CP, KP and LEL are members of DZIF. AH is supported by the BMBF [01KA1611, 01KA2027 and 01KA2113A] and additionally funded by the Deutsche Forschungsgemeinschaft (DFG) under Germany's Excellence Strategy – EXC2151 – 390873048. DM received financial support from the DFG [Project-ID 398967434 – TRR 261. The funders had no role in study design, data collection and analysis, decision to publish, or preparation of the manuscript.

**Competing interests:** The authors have declared that no competing interests exist.

infection. Furthermore, highest levels of NATOG were found in *O. volvulus* positive nodding syndrome patients. These findings highlight that the detection of distinct biomarkers might be useful for a differential diagnosis of epilepsy and nodding syndrome in *O. volvulus* endemic areas.

**Trial-registration:** NCT03653975.

## Author summary

A lot of epidemiological studies suggest that an infection with the filarial nematode *Onchocerca volvulus* is a risk factor for a certain form of epilepsy which includes the nodding syndrome. The seizures occur suddenly in previously healthy children between the ages of 3 and 18 years and the disease is an important public health problem in onchocerciasis-endemic regions with inadequate or lacking onchocerciasis elimination programmes. So far, no exact pathophysiological mechanisms were detected that shed light on how the parasite can trigger the seizures, although many factors have been investigated. However, most studies lack an accurate determination of an active/chronic *O. volvulus* infection; rather, they rely on the presence of circulating antibodies, which is only evidence of previous exposure to the parasite (and therefore positive in most people living in endemic areas. Within our study, we included participants from the Mahenge area, Tanzania, who were displaying epilepsy/nodding syndrome in absence or presence of a PCR-determined *O. volvulus* infection. We could show that the group of *O. volvulus* infected patients displayed higher levels of eosinophils, antigen-specific immunoglobulin levels as well as increased levels of a worm-related metabolic product (NATOG), which might be used as a potential diagnostic biomarker of the disease in future studies.

## Introduction

Infections with the filarial parasite *Onchocerca volvulus* can cause river blindness. The disease is not usually induced by adult worms, but rather elicited by the death of their offspring, the microfilariae (MF), which pass through the skin and cornea (causing dermatitis with depigmentation and itching and visual impairment leading to blindness in a minority of infected persons) [1]. However, there is increasing evidence that infection with filariae may also be associated with epilepsy which is why the term onchocerciasis-associated epilepsy (OAE) has been introduced. OAE seems to start without any obvious cause, in previously healthy children, between the ages of 3–18 years. Cases are reported in areas of high ongoing or past onchocerciasis transmission in various regions of sub-Saharan Africa including Uganda, Tanzania, the Democratic Republic of the Congo, the Central African Republic, Cameroon and South Sudan [2–5]. OAE has public health implications in endemic regions; with more than 381,000 people estimated to be affected in 2015 [6]. Nodding syndrome, an epilepsy disorder that is characterised by head nodding seizures (repeated, involuntary forward bobbing), has also been associated with the presence of *O. volvulus* infections [4,7–10]. The disease may be accompanied by stunting, wasting, physical deformities, delayed sexual development, and psychiatric disorders in addition to neurological deterioration [11–14]. Nodding syndrome has been discussed to be associated with genetic traits of the parasite, distinct types of *O. volvulus* parasites (i.e., savannah versus forest), or a genetic contribution of the affected patients themselves [12,15,16]. So far, there is no available biological diagnostic test for identification of

early signs/symptoms that would enable timely treatment of neurological disorders of affected individuals [7].

Although an association between infections with *O. volvulus* and epilepsy has been reported since the 1930s, the pathophysiological pathway(s) involved in this manifestation remain unclear. Treatment studies with ivermectin (IVM), a medication typically used to combat the transmission stage (MF) of *O. volvulus*, have shown decreased incidence rates in new onset epilepsy cases in *O. volvulus* endemic areas [17–20]. In addition, an association between seizure severity and microfilarial density has been described [21]. However, MF are rarely detected in the cerebrospinal fluid of affected individuals and they have not been shown to penetrate the blood–brain barrier [22–24] but this might not exclude the crossing of worm proteins. In post-mortem studies, neuropathological examination of brains from people suffering from OAE reported the presence of accumulated tau proteins which are a marker of neurodegeneration [25,26], whereas another study postulated that a neuroinflammation rather than a tauopathy is the cause of OAE [23]. Additionally, antibodies against leiomodin-1, an actin-binding protein mainly present in human muscle but also in human neurons, have been implicated in the development of autoimmune neuropathology since those antibodies display cross-reactivity with *O. volvulus* antigens, more precisely with tropomyosin and tropomodulin. This has been supported by the fact that serum samples of patients with nodding syndrome positive for leiomodin-1 antibodies were shown to have neurotoxic properties to neuronal cells *in vitro*, however, those autoantibodies were also present in healthy village controls, although to a lesser extent [27,28]. Another study in patients with OAE did not find any evidence for the involvement of leiomodin-1 antibodies in the pathogenesis of OAE [29], therefore, other immunological factors might also contribute to the development of epilepsy/nodding syndrome in *O. volvulus* endemic areas. Asymptomatic *O. volvulus*-infected individuals have dominant type 2 immune, regulatory and anti-inflammatory responses [30–34]. However, hyperreactive onchocerciasis patients, showing severe pathology, are characterized by an accentuated Th17/Th2 phenotype in comparisons to individuals with a generalised phenotype with no or minimal pathology [35]. Since OAE was linked to neuroinflammation [23], immune profiling of patients with or without *O. volvulus* infection who suffer from epilepsy/nodding syndrome needs to be performed to elucidate their causes.

To gain insights into the pathophysiology of nodding syndrome and other epilepsy disorders that occur in *O. volvulus* endemic areas, we present detailed immunological data of our study performed between 2014 and 2015 in the Mahenge area of Tanzania. Results were compared between nodding syndrome patients and those with other epilepsy disorders either with or without an *O. volvulus* infection determined via PCR. In addition, immunological parameters of *O. volvulus* negative individuals who were free of neurological disorders from the same endemic region were studied with regards to both neurological and immunological factors.

## Material and methods

### Ethics statement

Written informed consent was obtained from all individuals; for children and for patients who were unable to give consent, e.g., because of cognitive impairment, written informed consent was obtained by a parent or legal guardian. Ethical approval was provided by the University of Heidelberg (project number S-2018/2014), the University of Tübingen (project number 483/2014BO1), the University of Bonn (project number 261114), the Technical University of Munich (project number 285/14), Germany and the Muhimbili University of Health and Allied Sciences (MUHAS, Ref. No. 2014-10-06/AEC/Vol.IX/12), Tanzania. The trial is registered at ClinicalTrials.gov (Identifier: NCT03653975).

## Study setting and data collection

The study was performed in the Mahenge area, Ulanga district, in south eastern Tanzania between November 2014 and April 2015. The majority of the included individuals with epilepsy/nodding syndrome were already known patients at the Mahenge Epilepsy Clinic or at surrounding public health facilities where they received their anti-seizure medications monthly. Furthermore, by use of local radio announcements and involving key local government workers, potential patients with nodding syndrome were encouraged to attend the Mahenge Epilepsy Clinic on specific dates for further evaluation. Before inclusion in the study, diagnosis of epilepsy and nodding syndrome was reviewed by TW (neuropaediatrician) and JK (neurology resident): Diagnosis of epilepsy was made according to the operational clinical definition of epilepsy of the International League Against Epilepsy [36], based on clinical information, including interictal routine-EEG in all cases that were willing to have one performed (95/106). Individuals with nodding syndrome had to fulfil the diagnostic criteria of probable or confirmed nodding syndrome according to the WHO epidemiologic surveillance case definition for study enrolment [37].

To collect clinical data, a standardised questionnaire was used, which was approved by supervising neurologists (WM, ASW, ES). It was written in English and translated in both Kiswahili, a widely spoken national language and into Kipogoro, the local language in the Mahenge area. Local nurses supported the researcher team in informing the individuals about the study by translating and back translating, if any questions arose regarding informed consent. Patients with co-infections such as HIV and TB, concomitant cardiovascular, renal, immunologic, or hematologic diseases based on current/past medical history and clinical examination were excluded from the present study and directed to specific health facilities for treatment.

The study site has had a high endemicity of onchocerciasis since the early last century [38] and is among the main transmission foci in Tanzania. For serological-parasitological analysis, blood, urine and skin samples were ascertained at the Mahenge Epilepsy Clinic. Skin biopsies were taken at the iliac crest (two per individual) as previously described [33] and were immediately investigated microscopically for the presence of *O. volvulus*-specific MF. Skin snips were stored in isopropanol and then transported to Bonn, Germany, where a specific real-time PCR was performed to determine the presence of *O. volvulus* and/or *Wolbachia* endobacteria infection (OvwFtsZ/actin duplex real-time PCR, performed by CL) [39]. Thus, our cohort consisted of patients presenting with epilepsy/nodding syndrome who were *O. volvulus* positive (Ov⁺EpNd) or negative (EpNd) by PCR. In addition, volunteer relatives and caregivers of patients without neurological signs/symptoms were recruited from the waiting area of the Mahenge Epilepsy Clinic. Furthermore, village controls from the same villages that patients came from were also included as not every patient came with a suitable and consenting volunteer relative to the Mahenge Epilepsy Clinic. We were unable to perform age and sex matching at the individual level, but the median age and sex distribution was not significantly different between the groups (Table 1). Plasma and urine samples were collected at the Mahenge Epilepsy Clinic and stored at -20°C until further use in Bonn, Germany. Neurocysticercosis was not assessed as at that time there was no pig smallholder business in the Mahenge area. Moreover, neurocysticercosis is characterized by a later onset of epilepsy with people usually experiencing their first seizure after the age of 20 years [40].

## Assessment of immune cell composition in peripheral whole blood

Blood smears were screened for their cellular compositions. Samples were fixed on microscope slides using methanol and stained following immersion in a standard Giemsa stain solution

**Table 1. Characteristics of study cohort.**

|  | CTRL (n = 38) | EpNd (n = 45) | OV⁺ EpNd (n = 61) | p value |
|---|---|---|---|---|
| **Median age [range]** | 25.5 years [13–54] | 21.5 years [7–38] | 21 years [7–50] | p = 0.732 |
| **Sex** | 50.0% males | 48.8% males | 42.6% males | p = 0.722 |
|  | 50.0% females | 51.2% females | 57.4% females |  |
| **Individuals with seizures in last 12 months** | 0% | 73.7% | 71.9% | p = 0.851 |
| **MF⁺ individuals** | 0% | 0% | 55.7% | p< 0.001 |
| **MF/g of skin [mean, range]** | 0 | 0 | 7.48 [0.50–114.2] |  |

According to their diagnostic status, individuals were categorized as CTRL (without *O. volvulus* infection and epilepsy disorder or nodding syndrome), EpNd (with epilepsy disorders or nodding syndrome) or OV⁺ EpNd (*O. volvulus* infected presenting epilepsy disorders or nodding syndrome). p values denote statistical differences between the groups for that parameter tested by chi square test.

(Merck KGaA, Darmstadt, Germany) for 20 minutes. The cellular composition was determined after classifying 100 immune cells per individual under a 100x magnification with an Axioscope microscope (Zeiss, Oberkochen, Germany) in a blind manner to avoid bias. Immune cells included macrophages/monocytes, lymphocytes, neutrophils and eosinophils.

## Assessment of cytokines, immunoglobulins, neurodegeneration markers, chemokines and leiomodin-1

Plasma samples from included participants were analysed for the content of IFN-γ (detection limit 4 pg/ml), IL-6 (detection limit 2 pg/ml), IL-10 (detection limit 2 pg/ml), total IgG4 (detection limit 31.3 ng/ml), total IgE (detection limit 7.8 ng/ml) (Thermo Fisher Scientific, Bender MedSystems GmBH, Vienna, Austria) and eosinophil cationic protein (ECP, detection limit 30 pg/ml, Aviscera Bioscience, Inc, Santa Clara, USA) using human ELISA kits according to the manufacturer's instructions. Leiomodin-1 levels (detection limit 0.295 ng/ml) were measured using the Human Leiomodin-1 ELISA Kit from EIAab, Wuhan, China in accordance with the manufacturer's instructions. Levels of *O. volvulus*-specific IgE and IgG1-4 were also measured as described previously [33]. All plates were measured using a SpectraMAX ELISA reader (Molecular Devices, Sunnyvale, USA) with wavelength correction (450 nm and 570 nm) and analysed with SOFTmax Pro 3.0 software. In addition, chemokine levels and neurodegeneration markers were analysed via Luminex technology using the Chemokine 9-Plex Human ProcartaPlex Panel 1 and the 9-plex ProcartaPlex Human Neurodegeneration Panel 1 kit (Thermo Fisher Scientific, Schwerte, Germany), respectively. The limits (upper limit of quantification/lower limit of quantification) of the chemokines and neurodegeneration markers in pg/ml were as follows: eotaxin (2,350/0.57), Gro-α (3,225/3.15), IL-8 (8,950/2.19), IP-10 (6,000/1.46), MCP-1 (3,850/3.76), MIP-1α (2,288/2.23), MIP-1β (7,050/6.88), RANTES (200/0.78), SDF-1α (14,475/14), amyloid-β 1–40 (1847,000/451), amyloid-β 1–42 (1,700/0.42), FGF-21 (35,400/8.64), kallikrein-6 (22,800/5.57), NCAM-1 (221,900/54), neurogranin (41,600/10), tau-pt181 (2000/1.95), tau total (80,000/20) and TDP-43 (393,200/96). Data were acquired using a MAGPIX Luminex system (Luminex Corporation, Austin, TX, USA) and analysed with ProcartaPlex Analyst software version 1.0 (Thermo Fisher Scientific).

## Determination of N-acetyltyramine-O, β-glucuronide (NATOG) levels in urine samples

Urine samples were investigated for NATOG levels as described recently [41]. In brief, removal of debris from thawed urine samples was accomplished by centrifugation at 19,000 x

g at room-temperature (RT) for 10 minutes using a FA-45-30-11 Eppendorf centrifuge (Eppendorf AG, Hamburg, Germany). A maximum of 100 μl of urine supernatant was transferred into a new 1 ml Eppendorf tube (Eppendorf AG) containing 400 μl ice-cold methanol (>99.92% MeOH, Merck, Darmstadt, Germany) and 20 μl of D3-NATOG (= 0.36 nmol); used as an internal standard. D3-NATOG was synthesised as described previously [41]. An additional centrifugation step at 19,000 x g for 10 minutes at RT was performed to precipitate proteins from the solution and the resulting supernatant was transferred into a new vial. Solutions were placed into a CHRIST AVC 2–18 speedvac (Martin Christ Gefriertrocknungsanlagen GmbH, Osterode, Germany) for at least 1.5 hours to dry the sample. Dried samples were reconstituted with 300 μl of ice-cold MeOH and centrifuged at 19,000 x g for 10 minutes at RT. Supernatants were transferred into glass inlets (VWR International GmbH, Darmstadt, Germany) and again placed in the CHRIST AVC 2–18 speedvac (30˚C for 1.5 hours at 210 x g RPM to remove solvents). Remaining sample material was dissolved in 100 μl water and MeOH (95:5) solution and stored in autosampler vials (VWR International GmbH, Darmstadt, Germany). Samples were then separated by liquid chromatography with a SUPELCOSIL ABZ+Plus column (Merck, Darmstadt, Germany) and analysed through a quadrupole time of flight capable Agilent 6530 Q-TOF mass spectrometer (Agilent, Santa Clara, USA) with accompanying electrospray ionization as described [41]. Measurements and qualitative analysis were assessed using the Agilent MassHunter software (Agilent). According to their retention times and mass spectra, standards of NATOG and D3-NATOG were injected to identify their standard peaks following fragmentation. NATOG and D3-NATOG were eluted at 19.5–20.5 minutes and measured after fragmentation (transition of m/z 356.135 to 121.065 and 359.157 to 121.065 respectively) [41].

### Statistical analyses

Statistical tests to compare different parameters between the indicated groups were performed on Prism software (GraphPad Prism, version 9.4.1). Before testing for statistical significances between the groups, we performed a D'Agostino-Person omnibus normality test to test the distribution of the values. Kruskal Wallis followed by Dunn's multiple comparisons tests were used to compare control individuals and people with epilepsy and those with nodding syndrome with or without an *O. volvulus* infection determined by PCR. For comparisons of continuous parameters, the Spearman correlation test was used. P-values of $\leq 0.05$ were considered significant.

## Results

### Study characteristics

In total, we recruited 187 individuals that initially participated in the study and provided skin snips for the performance of the *O. volvulus*-specific PCR. Out of these 187 individuals, only 144 provided blood samples of whom 106 cases suffered from epilepsy/nodding syndrome consisting of 61 individuals with *O. volvulus* infection (OV⁺EpNd) and 45 individuals without *O. volvulus* infection (EpNd). Additionally 38 controls who were *O. volvulus* PCR negative were included as depicted in Table 1. Since not all blood and urine samples were available, further analysis was continued with 144 individuals consisting of 25.53% children ($\leq 18$ years, 50% males, 50% females) and 74.47% adults (>19 years, 42.3% males, 57.7% females). The group of 144 individuals was composed of EpNd individuals who were *O. volvulus* positive (OV⁺EpNd, n = 61) or negative (EpNd, n = 45) by PCR, and 38 controls who were *O. volvulus* PCR negative (Table 1). Age and gender distribution was comparable between the three groups and the number of seizures within the last 12 months also did not differ between EpNd and

OV$^+$EpNd groups. The overall median of age at investigation was 22 years [range 7–54 years] and the median of age at onset of nodding syndrome or epilepsy was 9 years [range 2–46 years]. Most individuals worked as farmers (98/144) or still went to school (12/144). Information on current anti-seizure medication and history of ivermectin intake in the past five years (via WHO African Programme for Onchocerciasis Control) [42] are listed in S1 Table. The overall data set of the study population is shown in S4 Table.

### Elevated levels of eosinophils in OV$^+$EpNd individuals

Blood smear samples were investigated microscopically with regards to the cellular composition. Levels of monocytes (Fig 1A) or neutrophils (Fig 1B) were not significantly altered between the groups; however, the percentage of lymphocytes was decreased in both patient groups when compared to uninfected control individuals and this was significant in the OV$^+$EpNd individuals (Fig 1C). Moreover, the percentage of eosinophils was significantly increased in OV$^+$EpNd individuals when compared to either the CTRL or EpNd groups

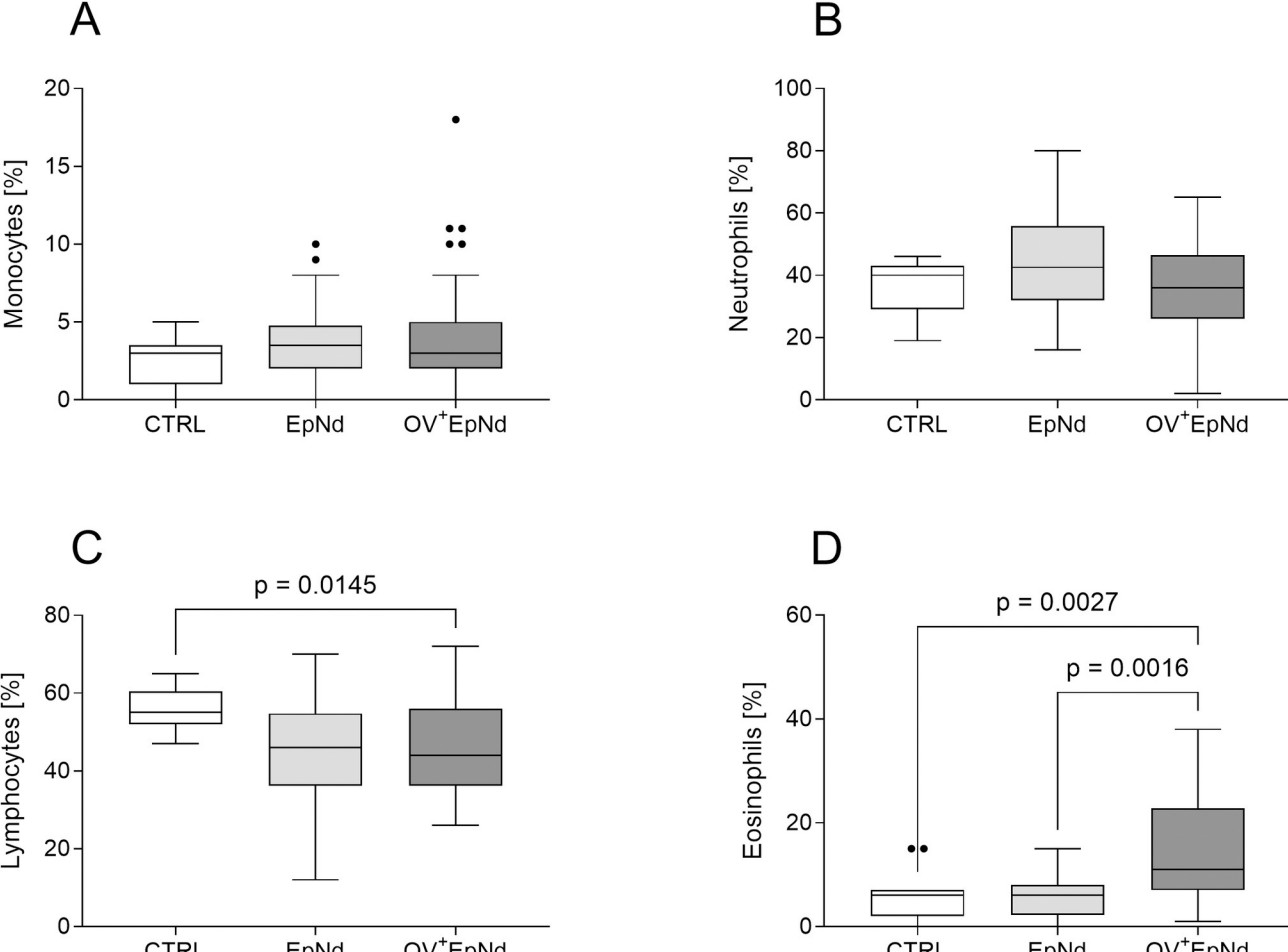

**Fig 1. OV$^+$EpNd individuals are characterized by elevated levels of eosinophils.** Blood smears from all patients were screened for their cellular composition and cells were counted for OV$^-$ individuals without neurological disease (CTRL, n = 15), those presenting epilepsy/nodding with (OV$^+$EpNd, n = 40) or those without an *O. volvulus* infection (EpNd, n = 24) at 100x magnification. Graphs show box whiskers with median, interquartile ranges and outliers of (A) monocytes, (B) neutrophils, (C) lymphocytes and (D) eosinophils. Statistical differences were obtained with Kruskal-Wallis followed by a Dunn's multiple comparison test.

(Fig 1D), suggesting that the immune cell composition in peripheral blood is changed in *O. volvulus* infected patients.

### Distinct chemokine profiles in OV⁺EpNd individuals

Since we observed that eosinophils (a common hallmark of helminth infections) were increased in OV⁺EpNd individuals (Fig 1D), we measured eosinophilic cationic protein (ECP), a ribonuclease superfamily member that has helminthotoxic properties [43]. Levels of ECP in plasma were decreased in EpNd individuals when compared to the other groups, and achieved significance when compared to OV⁺EpNd individuals (Fig 2A). In addition, levels of eosinophils correlated significantly with those ECP (Spearman-rho = 0.318, p = 0.013, all individuals included). Furthermore, we measured a panel of chemokines, including eotaxin, which are engaged in cell recruitment and function of immune cells. Interestingly, eotaxin was significantly lower in the OV⁺EpNd group when compared to CTRL individuals (Fig 2B), and the same was observed regarding SDF-1α levels (Fig 2C). However, levels of MIP-1α were significantly higher in the OV⁺EpNd individuals when compared to the CTRL group (Fig 2D). Individuals presenting EpNd alone showed significantly elevated levels of both MCP-1 (when compared to OV⁺EpNd individuals, Fig 2E) and Gro-α (in comparison to CTRL individuals, Fig 2F). Levels of IL-8, IP-10, MIP-1β and RANTES were also measured, but no significant differences were observed between the groups (S2 Table). These data highlight that the presence of an *O. volvulus*-infection in individuals displaying EpNd conditions alters their chemokine profile, indicating the potential for chemokines to be potential biomarkers for diagnosis.

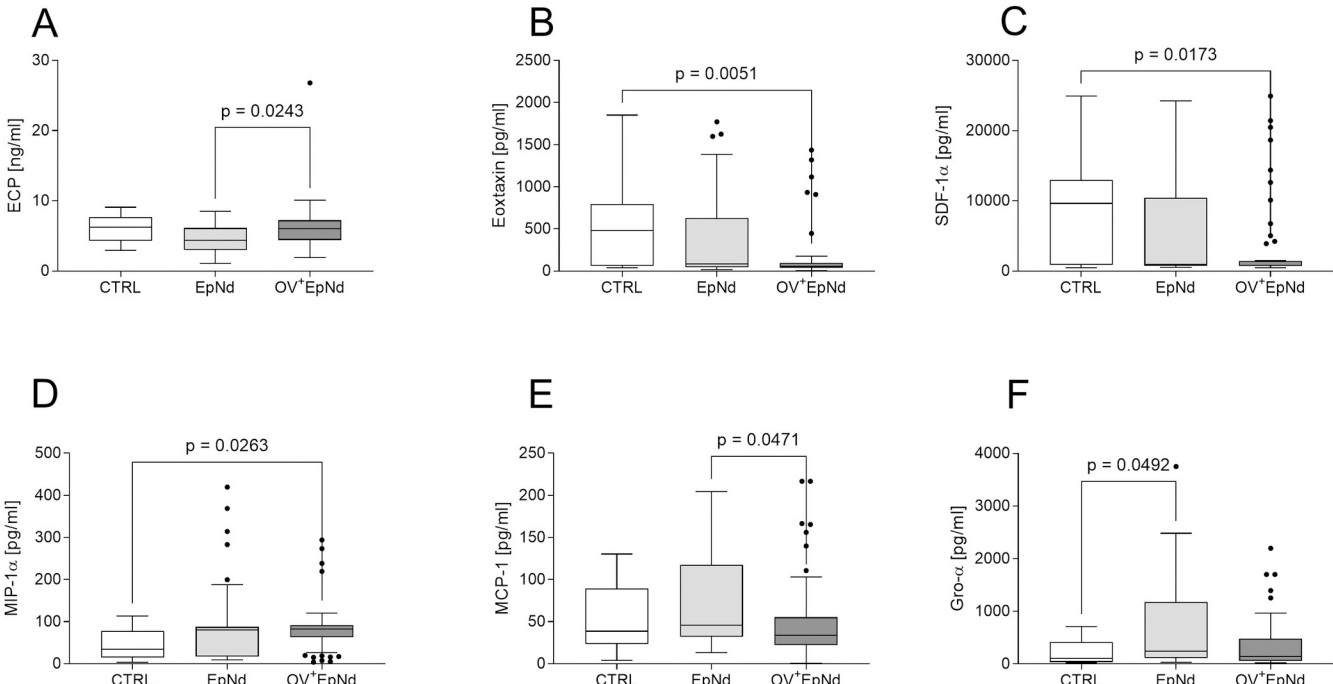

**Fig 2. Altered chemokine levels in OV⁺EpNd individuals.** Levels of (A) ECP from plasma samples (CTRL, n = 16, EpNd, n = 32, OV⁺EpNd, n = 51) were determined using ELISA. Levels of plasma-derived (B) eotaxin, (C) SDF-1α, (D) MIP-1α, (E) MCP-1 and (F) Gro-α were determined using Luminex technology (CTRL, n = 17, EpNd, n = 32, OV⁺EpNd, n = 54). Graphs show box whiskers with median, interquartile ranges and outliers. Statistical differences were obtained with Kruskal-Wallis followed by a Dunn's multiple comparison test.

### OV$^+$EpNd individuals are characterized by increased IL-10 and IL-6 levels

Next, we measured levels of IFN-γ, IL-10 and IL-6 in plasma samples. Whereas levels of IFN-γ were all under the detection limit, IL-10 and IL-6 levels were higher in both patient groups, but only significantly increased in OV$^+$EpNd individuals when compared to CTRL individuals (Fig 3A and 3B). The elevated IL-10 levels in the OV$^+$EpNd group correspond to similar findings in previous *O. volvulus*-related studies that were profiled in other endemic communities within Sub-Saharan Africa [33,35,44].

### Up-regulation of *O. volvulus*-specific Ig levels in OV$^+$EpNd individuals

IL-10 is a known inducer of IgG4 responses during *O. volvulus* infections [31,33]. Since we observed elevated levels of this anti-inflammatory cytokine, we also measured levels of *O. volvulus*-specific immunoglobulins in plasma samples of all three groups. As expected, levels of *O. volvulus*-specific IgG1-4 were significantly higher in the OV$^+$EpNd group when compared to CTRL and EpNd individuals (Fig 4A–4D). Moreover, when correlated with MF, it was revealed that all of the *O-volvulus*-specific IgG levels correlated significantly with the microfilarial load (MF/mg of skin, S3 Table). Neither *O. volvulus*-specific IgE nor the ratio of IgG4/IgE was significantly altered between the groups (Fig 4E and 4F). In addition, levels of total IgE and IgG4 were determined by commercially available ELISAs (S2A and S2B Fig). There was an increase in both parameters within the OV$^+$EpNd group and significantly higher levels of IgE were found in comparison to the CTRL group and the EpNd group (S2A Fig). The elevated *O. volvulus*-specific IgG levels in the OV$^+$EpNd group confirmed not only their infection state but, in association with the other markers, suggests a special immune profile for this patient group.

### Elevation of neurodegenerative markers in all individuals with epilepsy disorders

Next, we explored the potential association of neurodegenerative proteins in EpNd individuals and whether these were altered with concurrent *O. volvulus* infections. For this, we performed a Luminex assay that measured nine different neurodegenerative markers in plasma samples. When compared to CTRL, levels of amyloid-β 1–42 and TDP-43 were significantly elevated in

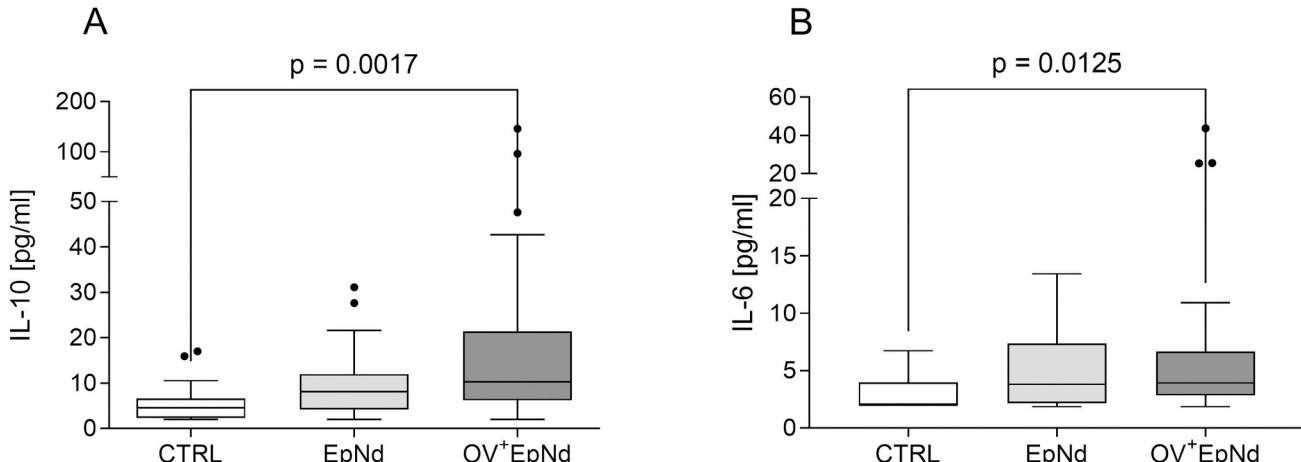

**Fig 3. Elevated IL-10 and IL-6 levels in OV$^+$EpNd patients.** Levels of (A) IL-10 and (B) IL-6 from plasma samples (CTRL, n = 16, EpNd, n = 32, OV$^+$EpNd, n = 51) were determined using ELISA. Graphs depict box whiskers with median, interquartile ranges and outliers. Statistical significances between the indicated groups were obtained with Kruskal-Wallis followed by a Dunn's multiple comparison test.

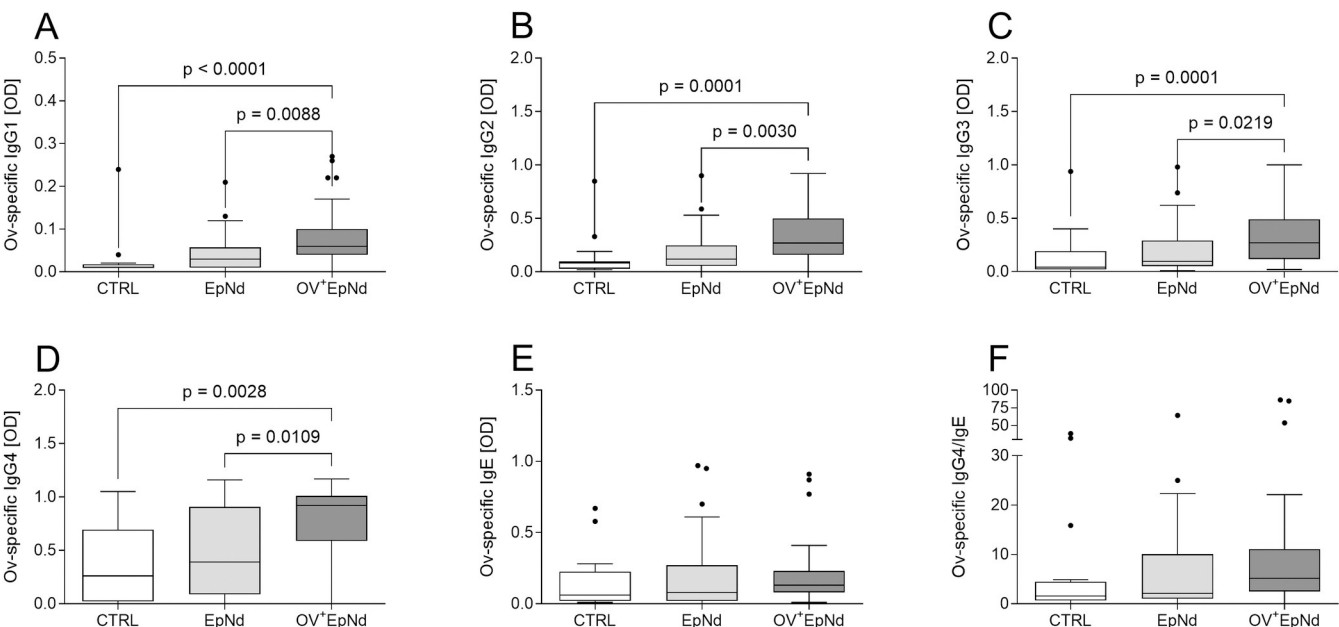

**Fig 4. Increased antigen-specific IgGs in OV⁺EpNd patients.** Levels of (A) *O. volvulus*-specific IgG1, (B) IgG2, (C) IgG3, (D) IgG4, (E) IgE and (F) the ratio of IgG4/IgE from plasma samples (CTRL, n = 16, EpNd, n = 32, OV⁺EpNd, n = 51) were determined using ELISA. Graphs depict box whiskers with median, interquartile ranges and outliers. Statistical significances between the indicated groups were obtained with Kruskal-Wallis followed by a Dunn's multiple comparison test.

both EpNd and OV⁺EpNd individuals (Fig 5A and 5B respectively). Neurogranin was increased in EpNd and OV⁺EpNd patients when compared to the CTRL individuals, albeit only significantly in the former group (Fig 5C). There were no significant differences detected between any of the three groups upon measuring amyloid-β 1–40, FGF-21, kallikrein-6, NCAM-1, tau total and tau pT181 (S2 Table). These findings show that these three neurodegenerative markers (amyloid-β 1–42, TDP-43 and neurogranin) are associated with seizures *per se* independent of an *O. volvulus* infection.

## Leiomodin-1 levels are reduced in EpNd patients with *O. volvulus* co-infection

Leiomodin-1 is a factor that is debated regarding its influence on OAE/nodding syndrome since antibodies against leiomodin-1 in the brain are cross-reactive against tropomyosin found

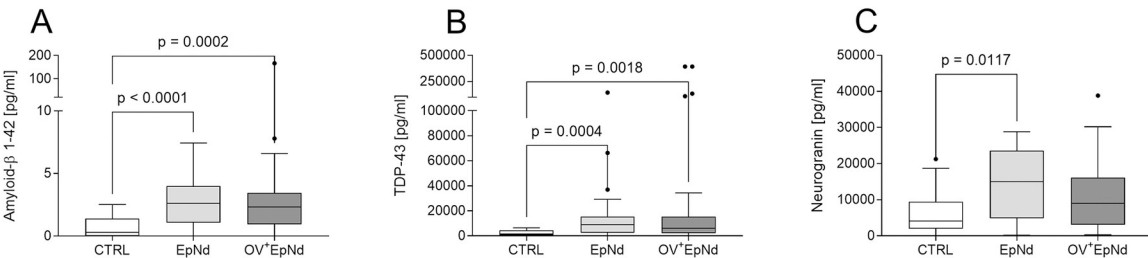

**Fig 5. Up-regulated neurodegeneration markers in patients with epilepsy/nodding syndrome.** Levels of neurodegeneration markers of included individuals from plasma samples (CTRL, n = 17, EpNd, n = 32, OV⁺EpNd, n = 54) were determined using Luminex technology. Data depict concentrations of (A) amyloid-β 1–42 (B) TDP-43 and (C) neurogranin. Graphs show box whiskers with median, interquartile ranges and outliers. Since data were non-parametric, statistical significances between the indicated groups were obtained with Kruskal-Wallis followed by a Dunn's multiple comparison test.

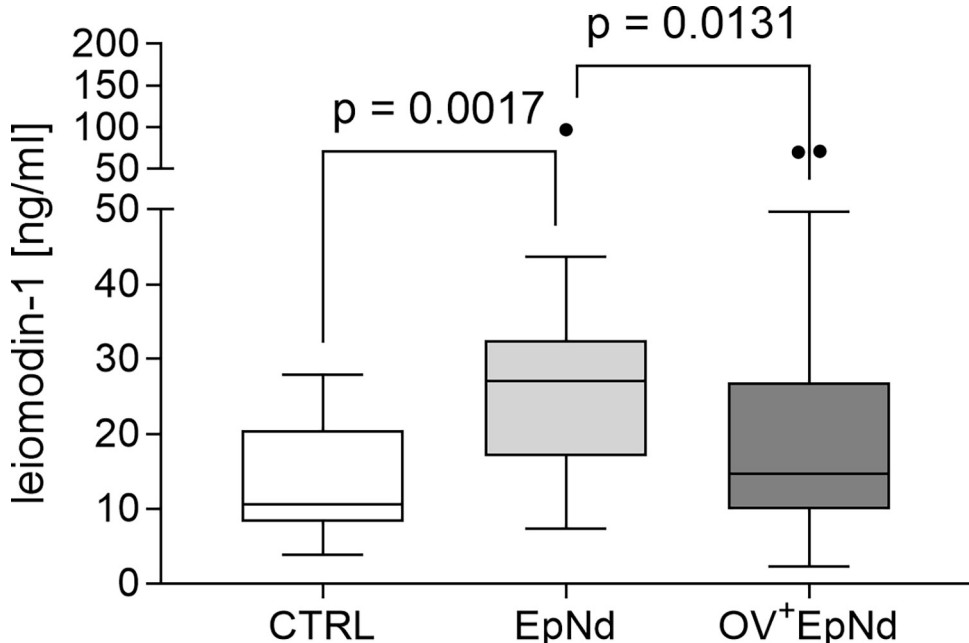

**Fig 6. Elevated levels of leiomodin-1 in EpNd patients.** Levels of leiomodin-1 were measured via ELISA from plasma samples (CTRL, n = 16, EpNd, n = 32, OV⁺EpNd, n = 51). Graph shows box whiskers with median, interquartile ranges and outliers. Since data were non-parametric, statistical significances between the indicated groups were obtained with Kruskal-Wallis followed by a Dunn's multiple comparison test.

in some *O. volvulus* infected individuals and may therefore induce neurotoxicity [27]. Levels of leiomodin-1 were determined via ELISA from all three investigated groups. However, significantly increased levels were only found in the EpNd cohort when compared to the OV⁺EpNd and CTRL groups (Fig 6).

## Increased urinary NATOG levels in OV⁺EpNd individuals and predominant in *O. volvulus* infected individuals presenting with nodding syndrome

N-acetyltyramine-O, β-glucuronide (NATOG) was previously described as a potential non-invasive biomarker of *O. volvulus* infections [45,46]. The metabolite NATOG results from the nematode-derived tyramine that is subsequently metabolised in the mammalian host, and has been observed to be significantly increased in *O. volvulus* positive patients with epilepsy compared to controls with or without *O. volvulus* infection [47]. We determined levels of NATOG in our Tanzanian cohort using liquid chromatography–mass spectrometry and found significantly increased levels of NATOG (median 4.61 µM, IQR: 1.1–17.75) in the OV⁺EpNd patients when compared to CTRL individuals (median 0.35 µM, IQR: 0.15–0.923) as well as compared to the EpNd group cohort (median 1.18, IQR: 0.30–3.79), albeit not significantly (Fig 7A). Moreover, levels correlated slightly with MF counts (Spearman-rho = 0.285, p = 0.026) and stronger with the OV copies/µl *per se* (Spearman-rho = 0.521, p<0.001). Finally, we determined the levels of NATOG in individuals presenting with either epilepsy (Ep) or nodding syndrome (Nd) separately with or without *O. volvulus* infection (Fig 7B). Here, the median urinary NATOG concentration was significantly higher in the OV⁺Nd group (median 8.90 µM, IQR: 3.60–25.24) when compared to either the CTRL group (median 0.35 µM, IQR: 0.15–0.923) or individuals with epilepsy without concomitant *O. volvulus* infection (median 0.57,

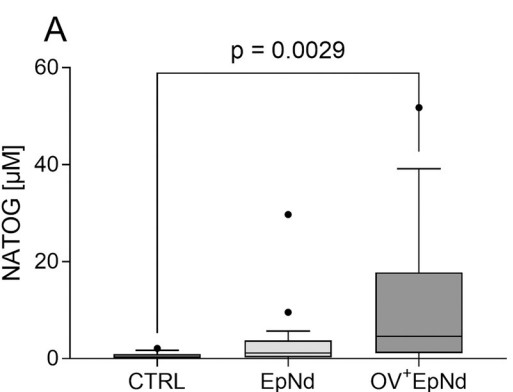 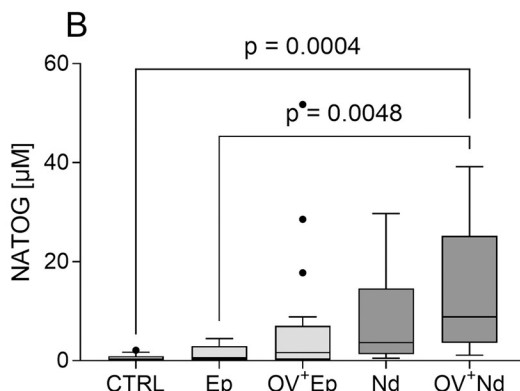

**Fig 7. Increased NATOG levels in patients with nodding syndrome.** Levels of NATOG from included individuals from urine samples were determined using liquid chromatography mass spectrometry. NATOG levels of individuals without neurological diseases with and without an *O. volvulus* infection (CTRL, n = 12, EpNd, n = 18, OV+EpNd, n = 31) (A), individuals are further subdivided into those with epilepsy and nodding syndrome with or without an *O. volvulus* infection (CTRL, n = 12, Ep, n = 12, OV+EpNd, n = 19, Nd, n = 6, OV+Nd, n = 12) (B). Graphs show box whiskers with median, interquartile ranges and outliers. Since data were non-parametric, statistical significances between the indicated groups were obtained with Kruskal-Wallis followed by a Dunn's multiple comparison test.

IQR: 0.18–2.96), but not significantly increased with regards to OV+EP patients (median 1.65, IQR: 0.17–4.49). Thus, our findings show that urine-derived NATOG levels are increased in *O. volvulus* positive patients with nodding syndrome.

## Discussion

Although several epidemiological studies have shown a potential association between epilepsy and infection with the filarial nematode *O. volvulus* in different African countries [2–5], the mechanisms inducing pathogenesis are still unknown. Within our study, we aimed to investigate different immune parameters, neurodegeneration markers and the *O. volvulus* biomarker NATOG in patients from endemic areas presenting with nodding syndrome or other epilepsy disorders with or without a concomitant *O. volvulus* infection, determined via PCR of skin snips. These measured parameters were additionally compared to those of *O. volvulus*-uninfected and epilepsy/nodding syndrome-free individuals. We observed that OV+EpNd and EpNd patients display altered immune cell populations in the peripheral blood and modified chemokine and cytokine expression in plasma compared to the control cohort. In contrast, most of the measured parameters were not significantly altered between EpNd patients with or without concomitant *O. volvulus* infection. However, we could show that NATOG can differentiate between *O. volvulus* infected patients with nodding syndrome and those with epilepsy without filarial infection but not those with epilepsy with filarial infection.

Peripheral blood cells were already described to play a role during epileptic seizures [48]. With regards to neutrophils, we did not observe any alteration between the groups, whereas lymphocytes were significantly decreased in the peripheral blood of OV+EpNd patients. Low lymphocyte counts might also be the result of the *O. volvulus* infection *per se*, as untreated patients with onchocerciasis have been described to have lymphocytopenia [49]. Increased eosinophils are a hallmark of parasitic infections [50], and, indeed, we observed significantly higher percentages of eosinophils in our OV+EpNd group. This was in line with a previous publication from Winkler et al. investigating a population of nodding syndrome patients [12]. In contrast, the level of eosinophils in the *O. volvulus*-free EpNd were comparable to those of the control group.

Since we have seen significant differences in the cellular composition of our study partici-pants, we measured chemokines that are crucial for their recruitment and activity. The pro-inflammatory and helminthotoxic eosinophilic cationic protein [33,43] produced by eosino-phils was detected at lower levels in *O. volvulus* negative EpNd patients compared to the *O. volvulus* positive population. However, the OV[+]EpNd individuals did not display higher ECP concentrations than the control individuals, despite having significantly higher eosinophil counts. ECP is described to be up-regulated in Sowda patients who suffer from severe dermati-tis [51], in contrast, our *O. volvulus*-infected individuals did not show severe pathology, which might explain the unaltered production of ECP when compared to the control individuals.

Another important factor involved in the recruitment of eosinophils is eotaxin. In contrast to the higher percentages of eosinophils, we found decreased eotaxin levels in OV[+]EpNd indi-viduals. This indicates that, although OV[+]EpNd individuals are characterized by higher levels of blood eosinophils, these cells seem not to be functionally active, at least during the time point of our observation. It might be of interest to determine their activity status during or shortly after an epileptic seizure. Furthermore, we measured significantly reduced levels of SDF-1α (CXCL12) in the OV[+]EpNd cohort compared to the control individuals. SDF-1α is described as major player at the interface of the immune and the nervous system [52]; elevated in elderly patients with autoimmune or late onset epilepsy [53]. Again, it would be interesting to see how the SDF-1α levels are modulated during or after a seizure.

Levels of Gro-α of EpNd patients were also significantly elevated compared to the control individuals, while OV[+]EpNd individuals showed similar levels as the control cohort. Gro-α is usually involved in neutrophil chemotaxis and degranulation during inflammation [54]. Inter-estingly, it was shown that it is also involved during the development of human temporal lobe epilepsy [55]. Although levels of neutrophils were slightly increased in the EpNd individuals, their percentages did not correlate with the levels of Gro-α. Still, there might be a shift from neutrophils to eosinophils in *O. volvulus* infected patients. Interestingly, MCP-1 (CCL2), a cytokine involved in the recruitment of monocytes and other immune cells, i.e., neutrophils to inflammatory sites [56], was also increased in the EpNd cohort compared to OV[+]EpNd indi-viduals. In a previous study, higher levels of MCP-1 were found in brain tissue of patients with intractable epilepsy [57]. However, the findings are not comparable to the results of the study by Ogwang et al. [58], in which decreased concentrations of MCP-1 were detected in plasma samples of nodding syndrome individuals compared to healthy controls. The different out-come of the measurements could be due to differences in patient inclusion; their study included individuals with nodding syndrome based on the presence of OV-16 antibodies, a marker for exposure to *O. volvulus*, rather than examination and detailed history focused on nodding syndrome.

In addition, we found increased levels of MIP-1α in OV[+]EpNd patients. This chemokine is associated with the activation of inflammatory monocyte responses [59]. Although we observed a tendency of elevated levels of monocytes within both EpNd groups, they did not correlate with the chemokine. A recent pilot study by Vieri revealed no differences of MIP-1α levels between patients with epilepsy in *O. volvulus* endemic areas and control individuals [60]. However, in contrast to our study, they investigated cerebrospinal fluid samples and the defini-tion of cases was again based on the presence of OV-16 antibodies, which makes a direct com-parison of the results difficult.

Furthermore, the neurodegeneration markers amyloid-β 1–42, and TDP-43 were signifi-cantly elevated in OV[+]EpNd and EpNd individuals when compared to the control partici-pants. In contrast, a previous study from Hotterbeekx did not detect a signal for TDP-43 with immunohistochemistry of post-mortem brain samples from individuals with nodding syn-drome and OAE [23]. Higher levels of neurogranin were also seen in both groups presenting

with nodding syndrome and other epilepsy disorders in our study population. This observation is in line with a recent publication that described that neurogranin can be used as a biomarker for epilepsy since serum levels of people with epilepsy were higher in comparison to those of healthy individuals [61]. Taken together, our data indicate that these three neurodegeneration markers are independent of an *O. volvulus* infection and more related to the occurrence of seizures *per se*.

Individuals with a generalized form of the *O. volvulus* infection are characterized by high levels of IL-10 [31,33,35,44]. Indeed, we observed increased levels of IL-10 in OV+EpNd individuals compared to the control cohort. However, levels of IL-6 were also increased in both patient populations when compared to the control individuals, and were significant for the OV+EpNd patients (EpNd versus CTRL p = 0.077) in line with previous findings connecting elevated IL-6 plasma levels to temporal lobe epilepsy [62,63].

Consistent with elevated IL-10 levels, increased *O. volvulus* IgG4 levels are also a hallmark of individuals with the generalized form of the *O. volvulus* infection [31,35,44]. Whereas antigen-specific IgE levels were not altered, we detected the highest IgG4 levels within the OV+EpNd group again confirming that participants did not suffer from severe *O. volvulus* associated pathology like dermatitis. Interestingly, levels of *O. volvulus*-specific IgG1-3 were significantly increased when compared to the other groups, which confirmed our PCR data and subsequent classification of patients. Since we have seen additional correlations of the MF load with the different IgGs, future studies should investigate the potential differences between MF+ and amicrofilaridermic individuals with nodding syndrome and other epilepsy disorders in more detail.

Antibodies against specific *O. volvulus* antigens, i.e. tropomyosin and tropomodulin, were shown to be cross-reactive with leiomodin-1, which has recently been demonstrated to be present in human neurons and has been implicated in neurotoxicity in patients with nodding syndrome infected with *O. volvulus* [27]. It is still unclear how these cross-reactive antibodies are involved in the development of epilepsy-related disorders in onchocerciasis regions. A previous study by Hotterbeekx et al. [29] performed in the Democratic Republic of Congo and South Sudan was not able to confirm the presence of leiomodin-1 in the brain tissue of individuals with OAE nor in serum or cerebrospinal fluid samples. In our cohort, we observed significantly increased levels of leiomodin-1 in EpNd individuals when compared to OV+EpNd indicating no link of this parameter and the parasitic infection. The comparison of the different studies is difficult since, for example, people with nodding seizures are described to have a more severe epilepsy and higher microfilarial loads compared to people with other forms of onchocerciasis associated epilepsy [21,29]. In addition, in our study we only had leiomodin data from 10 MF+ individuals with nodding syndrome. Further studies with more participants are needed to investigate this connection.

Finally, we investigated the *O. volvulus* biomarker NATOG from urine samples of our participants. Another study could show that the mean NATOG content was elevated in urine of *O. volvulus*-infected individuals compared to non-infected individuals, however, NATOG levels could vary strongly [41]. Hotterbeekx and colleagues recently showed an association of NATOG levels in the urine of *O. volvulus*-positive individuals (OV-16 RDT-positive antibodies and/or MF) from the Democratic Republic of Congo [24]. In our cohort, we were able to observe significantly higher NATOG levels in OV+EpNd patients when compared to the control group. Additionally, the divided subpopulations had significantly elevated NATOG levels in OV+ Nd individuals. Thus, future studies with increased patient sample size within the nodding syndrome group could confirm if NATOG is a potential biomarker for nodding syndrome.

In summary, this study investigated the potential of *O. volvulus* modulated immune profiles of individuals suffering from epilepsy in O. *volvulus* endemic areas in comparison to individuals without neurological disorders. Although some significant distinctions within the

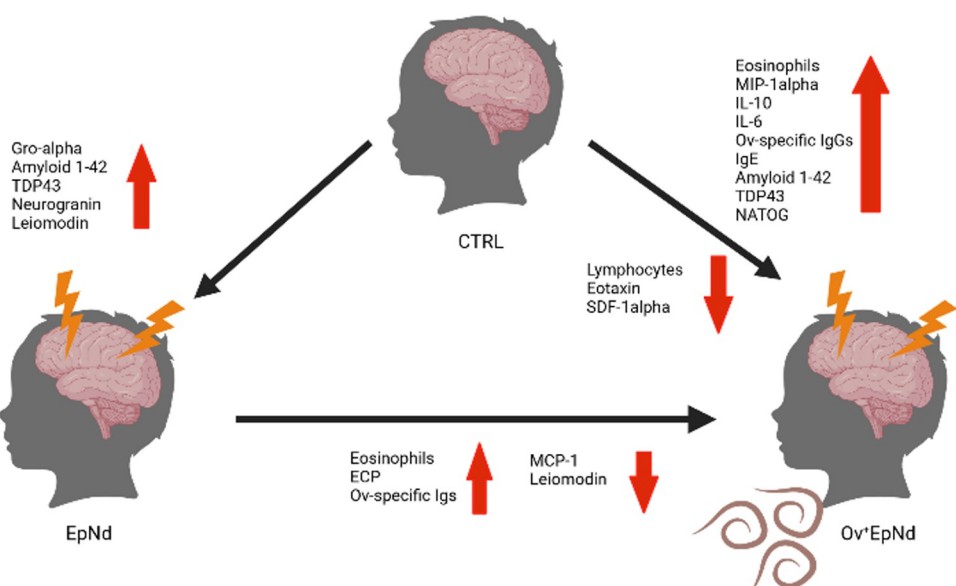

**Fig 8. Summary of distinct immune profiles of individuals in areas endemic for *O. volvulus* infection.** Individuals presenting no neurological pathology and *O. volvulus* infection were compared to patients with epilepsy/nodding syndrome with (OV⁺EpNd) or without concomitant filarial infection (EpND). Created with BioRender.com.

measurements of cytokine and neurodegeneration markers were observed between the OV⁺ and OV⁻ groups and controls (summarized in Fig 8), these did not allow straightforward conclusions which might be the result of the low number of participants within each group. We did not subdivide our OV⁺EpNd cohort according to presence or absence of MF since the group numbers for individual analysed parameters were too low. However, in future studies, analysis of OV⁺EpNd individuals should also include the comparison of MF status. Additionally, Th2 cytokines, as markers for helminthic infections should be integrated in these future studies which was not possible in our study due to the limitation of the plasma volume. It should be noted that recruitment of patients and collection of their samples was not performed at specific time points (e.g., shortly after or during an epileptic seizure) and the discrimination of patients with nodding syndrome and onchocerciasis associated epilepsy is difficult, relying on questionnaires (e.g., on the information of relatives) in remote areas. Therefore, more investigations are needed which take into account the time points of the last seizure and their frequencies. Moreover, future studies would need to measure factors specifically expressed in cerebrospinal fluid as well as the inclusion of an additional group of individuals without neurological disorders but with an active *O. volvulus* infection. Measurements of NATOG from urine samples suggests that the OV⁺EpNd group can be differentiated from the controls. A simplified version of this method, for example as a rapid antibody test, could support diagnostic methods in the field, particularly in children that are reluctant to undergo invasive methods. NATOG levels were even more pronounced in OV⁺Nd⁺ patients than in OV⁺Ep⁺ patients (albeit not significantly). However, further studies with higher patient numbers are needed to clarify the potential of NATOG as a biomarker for nodding syndrome.

## Supporting information

**S1 STROBE Checklist. This study adheres to the Strobe guidelines for cross-sectional studies.**
(DOC)

**S1 Table. Diagnosis of nodding syndrome was made according to the WHO case definition [37], all individuals fulfilled diagnostic criteria of probable nodding syndrome, and 2 individuals were confirmed cases by ictal video-EEG monitoring.** Diagnosis of epilepsy was made according to the definition of epilepsy of the international league against epilepsy [36]. (XLSX)

**S2 Table. Overview about levels of neurodegeneration markers and chemokines.** (DOCX)

**S3 Table. Significant correlation of MF load and antigen-specific IgG1-IgG4 and total IgG4.** MF load was correlated with the optical density of *O. volvulus*-specific immunoglobulins determined by ELISA from all available plasma samples (n = 99). A Spearman correlation test was applied. (DOCX)

**S4 Table. Overall data set of the study population.** (XLSX)

**S1 Fig. Overview of study participants recruited from the Mahenge district of Tanzania.** (TIF)

**S2 Fig. OV$^+$EpNd are characterized by elevated IgE and IgG4 levels.** Levels of total (A) IgE and (B) total IgG4 from plasma samples (CTRL, n = 16, EpNd, n = 32, OV$^+$EpNd, n = 51) were measured using ELISA technology. Graphs show box whiskers with median, interquartile ranges and outliers. Statistical significances between the indicated groups were obtained with Kruskal-Wallis followed by a Dunn's multiple comparison test. (TIF)

## Acknowledgments

The authors would like to acknowledge the health workers and officials in the Ulanga district and all volunteers for participating into this study. Also, we want to thank Özlem Mutluer from the Institute of Medical Microbiology, Immunology and Parasitology in Bonn for her excellent technical assistance.

## Author Contributions

**Conceptualization:** Clarissa Prazeres da Costa, Achim Hoerauf, Andrea S. Winkler.

**Data curation:** Kathrin Arndts, Josua Kegele, Alain S. Massarani, Thomas Wagner, Kenneth Pfarr.

**Formal analysis:** Kathrin Arndts, Alain S. Massarani, Thomas Wagner, Christine Lämmer, Helga Peisker, Dirk Menche, Mazen Al-Bahra.

**Funding acquisition:** Clarissa Prazeres da Costa, Achim Hoerauf, Andrea S. Winkler.

**Investigation:** Kathrin Arndts, Josua Kegele, Alain S. Massarani, Manuel Ritter, Christine Lämmer, Helga Peisker, Mazen Al-Bahra, Laura E. Layland-Heni.

**Methodology:** Kathrin Arndts, Josua Kegele, Thomas Wagner, Peter Dörmann, Helga Peisker.

**Project administration:** Kenneth Pfarr, Clarissa Prazeres da Costa, Erich Schmutzhard, William Matuja, Achim Hoerauf, Laura E. Layland-Heni, Andrea S. Winkler.

**Resources:** Peter Dörmann, Dirk Menche, Mazen Al-Bahra, William Matuja.

**Software:** Peter Dörmann.

**Supervision:** Manuel Ritter, Kenneth Pfarr, Erich Schmutzhard, William Matuja, Achim Hoerauf, Laura E. Layland-Heni, Andrea S. Winkler.

**Validation:** Josua Kegele.

**Writing – original draft:** Kathrin Arndts, Josua Kegele, Manuel Ritter, Laura E. Layland-Heni, Andrea S. Winkler.

**Writing – review & editing:** Kathrin Arndts, Josua Kegele, Manuel Ritter, Kenneth Pfarr, Erich Schmutzhard, Achim Hoerauf, Laura E. Layland-Heni, Andrea S. Winkler.

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
