## [Decision Letter · Decision Letter 0]

29 Mar 2023

Dear Dr. Arndts,

Thank you very much for submitting your manuscript "Epilepsy and Nodding Syndrome in Association with an Onchocerca volvulus Infection Drive Distinct Immune Profile Patterns" for consideration at PLOS Neglected Tropical Diseases. As with all papers reviewed by the journal, your manuscript was reviewed by members of the editorial board and by several independent reviewers. The reviewers appreciated the attention to an important topic. Based on the reviews, we are likely to accept this manuscript for publication, providing that you modify the manuscript according to the review recommendations. 

Please carefully evaluate the comments presented by reviewer 1, especially for small items such as "validated" survey, "most cases", etc. and revise the manuscript accordingly. This is a valuable piece of work and enhancing the clarity of your methods will make it an even more useful addition to the literature on this subject.

Sincerely,

Richard A. Bowen

Academic Editor

Cinzia Cantacessi

Section Editor

Please carefully evaluate the comments presented by reviewer 1, especially for small items such as "validated" survey, "most cases", etc. and revise the manuscript accordingly. This is a valuable piece of work and enhancing the clarity of your methods will make it an even more useful addition to the literature on this subject.

Reviewer's Responses to Questions

**Key Review Criteria Required for Acceptance?**

**Methods**

-Are the objectives of the study clearly articulated with a clear testable hypothesis stated?

-Is the study design appropriate to address the stated objectives?

-Is the population clearly described and appropriate for the hypothesis being tested?

-Is the sample size sufficient to ensure adequate power to address the hypothesis being tested?

-Were correct statistical analysis used to support conclusions?

-Are there concerns about ethical or regulatory requirements being met?

Reviewer #1: Major revisions: 

- Line 171: you state in the methodology section that the cases and controls were matched by age and sex. But then you never mention or describe this matching. For example, what is the accepted age range for matching? And how did you match 38 controls and 144 cases? 

Minor revisions:

- The ethics section might be more relevant at the end of the methodology section. We are better able to assess ethical considerations after having a clear description of the methodology.

- Line 142: meaning of regular care? Please specify "regular care" in the study area.

- Line 149: including routine interictal EEG in most cases. Please specify "most cases". 

- Line 152: Standardized and validated questionnaire. How was it validated?

- Line 157: concomitant cardiovascular and renal disease were excluded. Why? 

- No sample calculation. Why?

Reviewer #2: the objective of the study was clear. Authors investigated the immunological parameters in persons with NS and epilepsy with and without onchocerca infection. the population has been clearly described.

**Results**

-Does the analysis presented match the analysis plan?

-Are the results clearly and completely presented?

-Are the figures (Tables, Images) of sufficient quality for clarity?

Reviewer #1: Major revisions: 

- The representativeness of the sample was not easy to verify. A table describing the two groups with more variables other than sex and age might help us better appreciate the description of individuals, and potential confounding. The evolution/variation of immune cells and chemokines can be influenced by many reasons that we cannot attempt to appreciate even a little here

Reviewer #2: results are clearly stated and figures/tables are clear,

**Conclusions**

-Are the conclusions supported by the data presented?

-Are the limitations of analysis clearly described?

-Do the authors discuss how these data can be helpful to advance our understanding of the topic under study?

-Is public health relevance addressed?

Reviewer #1: conclusion is ok

Reviewer #2: (No Response)

**Editorial and Data Presentation Modifications?**

Reviewer #1: accept

Reviewer #2: (No Response)

**Summary and General Comments**

Reviewer #1: The authors present a study to investigate immunological parameters in individuals from Tanzania with epilepsy or nodding syndrome with or without O. volvulus infection and compared to negative individuals from the same endemic area without neurological disorders.

This study was original, and gives relevant data in the field of epilepsy and/or Nodding syndrome and the association with immunological disturbances.

The manuscript is well written and easy to read, except for the discussion section which is a bit long and dilutes the key messages.

To be published, some points need to be explained and addressed.

Reviewer #2: This is an important novel study that addresses the gap in knowledge of immune profile of persons with epilepsy or Nodding syndrome who are oncho infected. The study is well-designed and its minimal weaknesses are discussed by the authors adequately.

PLOS authors have the option to publish the peer review history of their article (what does this mean?). If published, this will include your full peer review and any attached files.

Reviewer #1: No

Reviewer #2: No

Figure Files:

Data Requirements:

Reproducibility:

References

---

## [Editor Report · Decision Letter 1]

8 Jun 2023

Dear Dr. Arndts,

Thank you very much for submitting your manuscript "Epilepsy and Nodding Syndrome in Association with an Onchocerca volvulus Infection Drive Distinct Immune Profile Patterns" for consideration at PLOS Neglected Tropical Diseases. As with all papers reviewed by the journal, your manuscript was reviewed by members of the editorial board and by several independent reviewers. The reviewers appreciated the attention to an important topic. Based on the reviews, we are likely to accept this manuscript for publication, providing that you modify the manuscript according to the review recommendations. 

Your manuscript has been reviewed carefully and there are a number of items that should be addressed from Reviewer 1. Please evaluate those comments, modify your manuscript for those items you believe could be improved and re-submit the modified manuscript along with comments about items you agree or disagree with.

Sincerely,

Richard A. Bowen

Academic Editor

Cinzia Cantacessi

Section Editor

Figure Files:

Data Requirements:

Reproducibility:

References

---

## [Editor Report · Decision Letter 2]

5 Jul 2023

Dear Dr. Arndts,

We are pleased to inform you that your manuscript 'Epilepsy and Nodding Syndrome in Association with an Onchocerca volvulus Infection Drive Distinct Immune Profile Patterns' has been provisionally accepted for publication in PLOS Neglected Tropical Diseases.

Best regards,

Richard A. Bowen

Academic Editor

Cinzia Cantacessi

Section Editor

Thank you for your patience during our review of this manuscript. One reviewer has some residual comments that you might want to look at but overall, we believe you have addressed all of the major points from earlier reviews and a decision of Accept has been delivered. Thank you for submitting to PLoS NTD.

---

## [Editor Report · Acceptance letter]

31 Jul 2023

Dear Dr. Arndts,

We are delighted to inform you that your manuscript, "Epilepsy and Nodding Syndrome in Association with an Onchocerca volvulus Infection Drive Distinct Immune Profile Patterns," has been formally accepted for publication in PLOS Neglected Tropical Diseases.

Best regards,

Shaden Kamhawi

co-Editor-in-Chief

Paul Brindley

co-Editor-in-Chief
